# $\alpha$-VAEs : Optimising variational inference by learning data-dependent divergence skew

**Jacob Deasy** [* 1]   **Tom McIver** [* 1]   **Nikola Simidjievski** [1]   **Pietro Liò** [1]

## Abstract

The *skew-geometric Jensen-Shannon divergence* $\left(\mathrm{JS}^{\mathrm{G}_\alpha}\right)$ allows for an intuitive interpolation between forward and reverse Kullback-Leibler (KL) divergence based on the skew parameter $\alpha$. While the benefits of the skew in $\mathrm{JS}^{\mathrm{G}_\alpha}$ are clear—balancing forward/reverse KL in a comprehensible manner—the choice of optimal skew remains opaque and requires an expensive grid search. In this paper we introduce $\alpha$-VAEs, which extend the $\mathrm{JS}^{\mathrm{G}_\alpha}$ variational autoencoder by allowing for learnable, and therefore data-dependent, skew. We motivate the use of a parameterised skew in the dual divergence by analysing trends dependent on data complexity in synthetic examples. We also prove and discuss the dependency of the divergence minimum on the input data and encoder parameters, before empirically demonstrating that this dependency does not reduce to either direction of KL divergence for benchmark datasets. Finally, we demonstrate that optimised skew values consistently converge across a range of initial values and provide improved denoising and reconstruction properties. These render $\alpha$-VAEs an efficient and practical modelling choice across a range of tasks, datasets, and domains.

## 1. Introduction

As variational inference (VI) progresses, state of the art Variational AutoEncoders (VAEs) increase in complexity (Vahdat & Kautz, 2020; Child, 2020) while continuing to maximise the Evidence Lower BOund (ELBO) (Blei et al., 2017). Compared to generative adversarial networks (GANs) (Goodfellow et al., 2014), VAEs necessitate less stringent and problem-dependent training regimes, and com-pared to autoregressive models (Larochelle & Murray, 2011; Germain et al., 2015) (that can be interpreted as instances of very deep VAEs (Child, 2020)) are less computation-ally expensive and more efficient to sample. VAE learning requires optimisation of an objective which balances the quality of decoded reconstructions from encoded representations, with a regularising divergence term penalising latent space deviations from a prior distribution.

To couple VAEs' learning with an appropriate regularisation, it is necessary to consider their underlying assumptions. VAEs often assume latent variables to be parameterised by a multivariate Gaussian, $p_\theta(z) = N(\mu, \sigma^2)$ with $z, \mu, \sigma \in \mathbb{R}^n$, approximated by $q_\phi(z|x)$ with $x \in \mathbb{R}^m$ and $n \leq m$. For instance, in the original VAE (Kingma & Welling, 2014), KL divergence (Kullback & Leibler, 1951) naturally constrained the variational distribution to an isotropic Gaussian unit ball KL $(q_\phi(z|x) \parallel \mathcal{N}(0, I))$, despite unfavourable properties (Bishop, 2006), such as un-boundedness and asymmetry. Moreover, KL does not cap-italise on the full flexibility of the wider family of expo-nential distributions, a recent direction which has tightened the ELBO (Brekelmans et al., 2020; Masrani et al., 2019) and rendered VAE divergence regularisation more inter-pretable in distribution space via *skew-geometric Jensen-Shannon* $\left(\mathrm{JS}^{\mathrm{G}_\alpha}\right)$ divergence (Nielsen, 2019; Deasy et al., 2020). We henceforth refer to VAEs with skewed geometric divergences as $\alpha$-VAEs, subsuming $\beta$-VAEs.

Divergence skew in VAEs balances the contrasting properties of forward and reverse KL (such as zero-avoidance/forcing) and circumvents opaque divergence terms by interpolating between them (see Chapter 10 in (Bishop, 2006)). However, an expensive grid search over skew values fixed through training is necessary to optimise for tasks such as image reconstruction. This is particularly problematic as the link between optimal skew and dataset properties is not clear and is not easily resolved. Moreover, when skew is treated as static, the divergence constraint does not change during training and therefore does not reflect improvements in the encoder and its embeddings. Instead, an improved optimisation would update the divergence skew relative to these factors without using prior knowledge or compromising performance.

---

[*]Equal contribution  [1]Department of Computer Science and Technology,University of Cambridge, United Kingdom. Correspondence to: Jacob Deasy <jd645@cam.ac.uk>.

Third workshop on *Invertible Neural Networks, Normalizing Flows, and Explicit Likelihood Models* (ICML 2021). Copyright 2021 by the author(s).

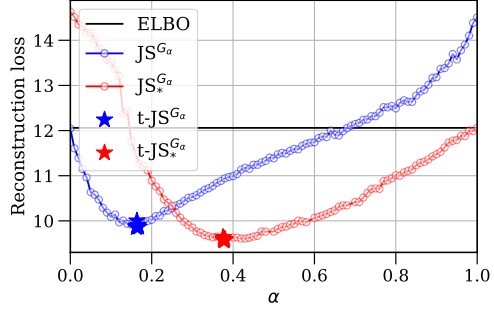

(a) $\alpha$-VAEs on the MNIST dataset.

| Divergence | KL expression | Train skew |
|---|---|---|
| KL (ELBO) | $\mathrm{KL}(q \parallel p)$ | ✗ |
| $\alpha$-VAEs | | |
| $\mathrm{JS}^{G_\alpha}$ | $(1-\alpha)\mathrm{KL}\left(p \parallel p^\alpha q^{1-\alpha}\right) + \alpha\mathrm{KL}\left(q \parallel p^\alpha q^{1-\alpha}\right)$ | ✗ |
| $\mathrm{JS}^{G_\alpha}_*$ | $(1-\alpha)\mathrm{KL}\left(p^\alpha q^{1-\alpha} \parallel p\right) + \alpha\mathrm{KL}\left(p^\alpha q^{1-\alpha} \parallel q\right)$ | ✗ |
| t-$\mathrm{JS}^{G_\alpha}$ | $(1-\alpha)\mathrm{KL}\left(p \parallel p^\alpha q^{1-\alpha}\right) + \alpha\mathrm{KL}\left(q \parallel p^\alpha q^{1-\alpha}\right)$ | ✓ |
| t-$\mathrm{JS}^{G_\alpha}_*$ | $(1-\alpha)\mathrm{KL}\left(p^\alpha q^{1-\alpha} \parallel p\right) + \alpha\mathrm{KL}\left(p^\alpha q^{1-\alpha} \parallel q\right)$ | ✓ |

(b) Breakdown of $\alpha$-VAE divergences.

*Figure 1.* The influence of divergence skew, $\alpha$, on test set image reconstruction (measured in mean squared error). Deasy et al. (2020) used an expensive grid search to fix divergence skew through training. Our separate optimisation of $\alpha$ in $\mathrm{JS}^{G_\alpha}$ (prefix t-), as a competing objective to reconstruction loss, leads to strong performance and substantially improves over reverse KL from any initial skew. We plot optimal $\alpha$, denoted by a star, for 5 training seeds which (visually) converge to the same point.

**Contributions.** To overcome these issues, we extend the work of Deasy et al. (2020) and introduce $\alpha$-VAEs, by allowing for data- and encoder-dependent skew. Our findings indicate trends in final skew values which are dependent on proxies for data complexity in synthetic examples. We demonstrate that optimal divergence skew tends toward a balance of forward and reverse KL as dimensionality increases and, for increasing distribution modes, training skew marginally favours forward KL. In the higher dimensional setting of standard image benchmark datasets, we then establish that final skew values converge across seeds and are consistent for a range of initial values. We further exhibit that learning skew in $\alpha$-VAEs has a positive impact on test set reconstruction loss (summarised in Figure 1), reconstructing denoised images from noisy images, and explain both advantages from the rate-distortion perspective (Alemi et al., 2018). Overall, we show that $\alpha$-VAEs with learnable $\alpha$ consistently outperform forward and reverse KL independent of dataset, encoder parameters, and initial skew [1].

## 2. The $\mathrm{JS}^{G_\alpha}$ divergence family

For distributions $P$ and $Q$ of a continuous random variable $Z = [Z_1, \ldots, Z_n]^{\mathrm{T}}$, the forward KL divergence (Kullback & Leibler, 1951) is defined as

$$\mathrm{KL}(P \parallel Q) = \int_Z p(z) \log\left[\frac{p(z)}{q(z)}\right] dz, \qquad (1)$$

where $p$ and $q$ are the probability densities of $P$ and $Q$ respectively, $z \in \mathbb{R}^n$, and reverse KL divergence refers to $\mathrm{KL}(Q \parallel P)$.

Reverse KL from a standard normal distribution $\mathcal{N}_2(0, I)$ to

a diagonal multivariate normal distribution $\mathcal{N}_1(\mu, \Sigma)$, $\mu \in \mathbb{R}^n$ and $\Sigma \in \mathbb{R}^{n \times n}$, is used throughout variational models (Higgins et al., 2017; Kingma & Welling, 2014; Neal, 2012) and is known to enforce zero-avoiding hyperparameters on $\mathcal{N}_1$ when minimised (Bishop, 2006; Murphy, 2012). On the other hand, the forward KL divergence is known for its zero-forcing property (Bishop, 2006; Murphy, 2012). However, there exist well-known drawbacks of the KL divergence, such as no upper bound leading to unstable optimisation and poor approximation (Hensman et al., 2014), as well as its asymmetric property $\mathrm{KL}(P \parallel Q) \neq \mathrm{KL}(Q \parallel P)$. Under-dispersed approximations relative to the exact posterior also produce difficulties with light-tailed posteriors when the variational distribution has heavier tails (Dieng et al., 2017).

One attempt at remedying these issues is the well-known symmetrisation, the Jensen-Shannon (JS) divergence (Lin, 1991)

$$\mathrm{JS}(p(z) \parallel q(z)) = \frac{1}{2}\mathrm{KL}\left(p \,\middle\|\, \frac{p+q}{2}\right) + \frac{1}{2}\mathrm{KL}\left(q \,\middle\|\, \frac{p+q}{2}\right). \qquad (2)$$

Although the JS divergence is bounded and offers some intuition through symmetry, it includes the problematic mixture distribution $\frac{p+q}{2}$. This term means that no closed-form expression exists for the JS divergence between two multivariate normal distributions using Equation (2).

Recently, (Nielsen, 2019) and (Nishiyama, 2018) have proposed a further generalisation of the JS divergence using abstract means (*quasi-arithmetic means* (Niculescu & Persson, 2006), also known as *Kolmogorov-Nagumo means*). By choosing the *weighted geometric mean* $\mathrm{G}_\alpha(x, y) = x^\alpha y^{1-\alpha}$ for $\alpha \in [0, 1]$, and using the property that the weighted product of exponential family distributions (which includes the multivariate normal) stays in the exponential family (Nielsen

---

[1]Code is available at:
https://github.com/jacobdeasy/geometric-js

& Garcia, 2009), we have the divergence

$$\mathrm{JS}^{G_\alpha}(p(z) \parallel q(z)) = (1 - \alpha)\mathrm{KL}\left(p \parallel G_\alpha(p, q)\right) \quad (3)$$
$$+ \alpha\mathrm{KL}\left(q \parallel G_\alpha(p, q)\right). \quad (4)$$

$\mathrm{JS}^{G_\alpha}$, the *skew-geometric Jensen-Shannon divergence*, between two multivariate Gaussians $\mathcal{N}_1(\mu_1, \Sigma_1)$ and $\mathcal{N}_2(\mu_2, \Sigma_2)$ then admits a closed form

$$\mathrm{JS}^{G_\alpha}\left(\mathcal{N}_1 \parallel \mathcal{N}_2\right) = (1 - \alpha)\mathrm{KL}\left(\mathcal{N}_1 \parallel \mathcal{N}_\alpha\right) \quad (5)$$
$$+ \alpha\mathrm{KL}\left(\mathcal{N}_2 \parallel \mathcal{N}_\alpha\right) \quad (6)$$

with the equivalent dual divergence being

$$\mathrm{JS}_*^{G_\alpha}\left(\mathcal{N}_1 \parallel \mathcal{N}_2\right) = (1 - \alpha)\mathrm{KL}\left(\mathcal{N}_\alpha \parallel \mathcal{N}_1\right) \quad (7)$$
$$+ \alpha\mathrm{KL}\left(\mathcal{N}_\alpha \parallel \mathcal{N}_2\right) \quad (8)$$

where $\mathcal{N}_\alpha$ has parameters

$$\Sigma_\alpha = \left((1 - \alpha)\Sigma_1^{-1} + \alpha\Sigma_2^{-1}\right)^{-1} \quad (9)$$
$$\mu_\alpha = \Sigma_\alpha \left((1 - \alpha)\Sigma_1^{-1}\mu_1 + \alpha\Sigma_2^{-1}\mu_2\right). \quad (10)$$

In simple terms, these divergences measure a weighted *arithmetic mean of divergences* from/to the prior/variational distribution to/from a weighted *geometric mean of distributions*. We proceed by examining why this skew parameter $\alpha$ should be learnt (replacing $\psi$ in Equation (14)), what this means in divergence space, and where the optimal $\alpha$ lie.

## 3. $\alpha$-VAE optimisation

### 3.1. Learning divergence parameterisations is not constrained optimisation

In order to flexibly interchange divergence terms regularising the latent space of VAEs, it is common to formulate VAE training as a constrained optimisation problem (Rezende & Viola, 2018; Higgins et al., 2017). A suitable objective to maximise is the marginal (log-)likelihood of the observed data $x \in \mathbb{R}^m$ as an expectation over the distribution of latent factors $z \in \mathbb{R}^n$

$$\max_\theta \left[\mathbb{E}_{p_\theta(z)}\left[p_\theta(x|z)\right]\right]. \quad (11)$$

The latent representation can be controlled by imposing an isotropic unit Gaussian constraint on the prior $p(z) = \mathcal{N}(0, I)$, arriving at the constrained optimisation problem

$$\max_{\phi,\theta} \mathbb{E}_{p_\mathcal{D}(x)}\left[\log \mathbb{E}_{q_\phi(z|x)}\left[p_\theta(x|z)\right]\right]$$
$$\text{subject to } D(q_\phi(z|x) \parallel p(z)) < \varepsilon, \quad (12)$$

where $\varepsilon$ dictates the strength of the constraint and $D$ is a divergence. Equation (12) is then rewritten as a Lagrangian

under the KKT conditions (Karush, 1939; Kuhn & Tucker, 2014), obtaining

$$\mathcal{F}(\theta, \phi, \lambda; x, z) = \mathbb{E}_{q_\phi(z|x)}\left[\log p_\theta(x|z)\right]$$
$$- \lambda\left(D(q_\phi(z|x) \parallel p(z)) - \varepsilon\right). \quad (13)$$

However, here we are interested in learning properties of the divergence itself, rendering Equation (12) and (13) invalid. In this setting, the constrained optimisation problem is no longer well-posed, as the constraint becomes part of the optimisation. Instead, we choose to relax these optimisation assumptions while maintaining competing objectives, the log-likelihood and the parameterised divergence

$$\mathcal{L}(\theta, \phi, \lambda; x, z) = \mathbb{E}_{q_\phi(z|x)}\left[\log p_\theta(x|z)\right]$$
$$- \lambda D_{\psi(x,z)}(q_\phi(z|x) \parallel p(z)), \quad (14)$$

where $\psi(x, z)$ parameterises $D$.

Such a formulation only relates back to Equation (13) in a valid manner when changes in $\psi$ map to a family of divergences and redundant cases are avoided. For instance, it would be particularly useful if, for a given $\psi$ during optimisation, training could still be understood as a reconstruction loss term and a closed-form divergence regularisation term. It is, therefore, of import to consider: which divergences allow for such a parameterisation, whether they will have non-trivial minima, and whether the subsequent properties of this optimisation generalise or are useful. To this end, we consider learning $\alpha$, the *skew*, in the $\mathrm{JS}^{G_\alpha}$ family of divergences (Appendix 2) and proceed by highlighting the connection to forward and reverse KL.

### 3.2. Skew optimisation

From an information geometric viewpoint, the skew parameter's influence on the intermediate distribution can be seen as the weighting of two distributions along a path in the statistical manifold. In (Masrani et al., 2019), the authors consider the geometric path between the variational distribution $q_\phi(z|x)$ and the model distribution $p(x, z)$. The expensive grid search therein required integration along this path and was later avoided in (Brekelmans et al., 2020) by using a moment-matching approach to learn the optimal point on the path. In contrast to these works, here we consider skew between $q_\phi(z|x)$ and the prior $p(z)$ as a form of regularisation.

Additionally, the property that $\mathrm{JS}^{G_\alpha}$ interpolates between forward and reverse KL (Deasy et al., 2020) offers a clear mechanism for trading off forward and reverse KL properties. In VAE optimisation, as we have generalised the overall objective by extending the constrained optimisation problem to competing objectives, it is possible to consider direct parameterisation of $\psi$ from Equation (14). As long as

$\psi$ maps to $[0, 1]$, the output is a valid divergence (a member of the $\mathrm{JS}^{G_\alpha}$ family) with extremes at forward and reverse KL, so we can consider optimising $\psi$ via gradient descent.

Before attempting the optimisation, we can derive useful properties of $\mathrm{JS}^{G_\alpha}(P \parallel Q)$ which clarify how the optimisation should be carried out. In particular, it is important to understand the behaviour of $\mathrm{JS}^{G_\alpha}$ with respect to $\alpha$, as this will dictate whether convergence is possible.

**Proposition 1.** *For Gaussian distributions $P$ and $Q$ with probability density functions $p(z)$ and $q_\phi(z|x)$ respectively, the derivative of $\mathrm{JS}^{G_\alpha}(P \parallel Q)$ with respect to $\alpha$ is*

$$\begin{aligned}
\frac{d\mathrm{JS}^{G_\alpha}}{d\alpha} = {} & 2(\alpha - 1)\mathrm{KL}(p(z) \parallel q_\phi(z|x)) \\
& + 2\alpha\mathrm{KL}(q_\phi(z|x) \parallel p(z)),
\end{aligned} \tag{15}$$

*with a stationary point at*

$$\alpha^* = \frac{\mathrm{KL}(p(z) \parallel q_\phi(z|x))}{\mathrm{KL}(p(z) \parallel q_\phi(z|x)) + \mathrm{KL}(q_\phi(z|x) \parallel p)}, \tag{16}$$

*which is a global minimum in the $\alpha$ dimension of the optimisation. Proof in Appendix B.1.*

A useful sanity check from Equation (16), is that $\alpha^*$ is clearly bounded between 0 and 1 due to the non-negativity of KL divergence. Secondly, as Equation (16) relies on parameters $\phi$, this expression is clearly data dependent, meaning optimal skew shifts during training and should not be fixed as in (Deasy et al., 2020). Finally, considered as a standalone $1D$ optimisation of $\alpha$, the global minimum means that training divergence skew is convex and should be simple to minimise in practice, with closed-form convergence.

**Proposition 2.** *The upper bound on the convergence rate of gradient descent to $\alpha^*$ in (16) is*

$$2\left(\mathrm{KL}(p(z) \parallel q_\phi(z|x)) + \mathrm{KL}(q_\phi(z|x) \parallel p(z))\right) D_1^2 e^{-4T}, \tag{17}$$

*where $D_1 = ||\alpha_1 - \alpha^*||_2^2$, $\alpha_1$ is the initial skew value, and $T$ is the number of optimisation steps. Proof in Appendix B.2.*

In the next section, we find practical optimisation of skew in $\mathrm{JS}^{G_\alpha}_*$ to be well behaved, suggesting that training skew in the dual divergence is also well-posed and does not reduce to an invalid divergence or one of the KL directions—despite an equivalent proof of convexity via differentiation not being obvious, due to a mixture term being outside of the log.

# 4. Experiments

As the convexity shown in Section 3.2 suggests simple optimisation of $\alpha$, we directly train $\alpha$ as another parameter of the model, but use a separate optimiser to the model parameters so that our 1D analysis holds.

## 4.1. Characterising skew optimisation for $\mathrm{JS}^{G_\alpha}_*$

To better understand how dual $\alpha$-VAEs, $\alpha_*$-VAEs, will behave in the more complex setting of modern variational inference benchmarks, we first highlight their properties on synthetic examples. In Figure 2, we depict different optimal skew values for a fit of a multivariate Gaussian to an underlying additive mixture of multivariate Gaussians with trained $\mathrm{JS}^{G_\alpha}_*$ skew. As the divergence integrals are not tractable, we directly optimise the multivariate Gaussian parameters via samples from the data for all divergences.

In both plots of Figure 2, we depict the emergent low-dimensional trends in optimised skew. For Figure 2a, we fit a 2D Gaussian ball to a 2D additive mixture of Gaussian balls, with an increasing number of components in the mixture. Whereas, in Figure 2b, we increase the dimension of the fit and keep the ratio of mixtures to dimensions fixed at 5. As the number of mixture components increases in Figure 2a, optimal skew for $\mathrm{JS}^{G_\alpha}_*$ decreases, favouring reverse KL as mass becomes more concentrated. Similarly, in Figure 2b, the optimal skew for $\mathrm{JS}^{G_\alpha}_*$ also decreases, underlining the need for data-dependent skewed divergences and suggesting learnt skew in dual $\alpha_*$-VAEs will be consistent and avoid trivial cases for more complex data.

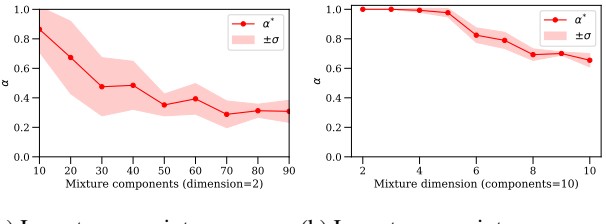

(a) Learnt $\alpha$ vs. mixture components.

(b) Learnt $\alpha$ vs. mixture dimension.

*Figure 2.* Emergent trends in optimised skew when fitting a Gaussian ball to an additive mixture of Gaussians in ND for increasing data complexity.

## 4.2. Benchmark image dataset performance

**Divergence skew convergence.** As the expression for the minimum in Equation (16) is encoder dependent, we begin our benchmark dataset assessment by depicting the alpha landscape. In Figure 3, we see that optimal $\alpha$ decreases during training, before stabilising in the final epochs. This supports the argument that naively fixing skew is not optimal, leads to inferior solutions, and even a hyperparameter search for fixed skew followed by training is not sufficient.

The divergences converge to a consistent minimum. This confirms that our $1D$ assessment of convexity and separate optimisation of $\alpha$ successfully leads to stable skew values (see Figure 4 in the Appendix). In addition, we exemplify how skew evolves across 5 different training seeds. These

| VAE | MNIST | Fashion-MNIST | dSprites | Chairs | CelebA |
|---|---|---|---|---|---|
| $\beta$-VAE (KL$(q \parallel p)$, $\beta = 4$, (Deasy et al., 2020)) | 11.75 | 13.32 | 10.51 | 20.79 | 269.52 |
| InfoVAE (MMD, $\lambda = 500$, (Deasy et al., 2020)) | 13.19 | 11.10 | 11.87 | 18.85 | 271.71 |
| Vanilla VAE (KL$(q \parallel p)$) | $10.67 \pm 0.27$ | $12.36 \pm 0.22$ | $7.78 \pm 0.24$ | $20.33 \pm 0.34$ | $262.53 \pm 2.07$ |
| $\alpha$-VAE (fixed $\alpha = 0.5$) | $11.24 \pm 0.05$ | $11.07 \pm 0.67$ | $12.07 \pm 0.06$ | $19.11 \pm 0.14$ | $270.33 \pm 0.78$ |
| $\alpha_*$-VAE (fixed $\alpha_* = 0.5$) | $8.82 \pm 0.04$ | $9.80 \pm 0.06$ | $5.72 \pm 0.07$ | $16.40 \pm 0.15$ | $264.27 \pm 0.45$ |
| $\alpha$-VAE | $8.89 \pm 0.07$ | $9.90 \pm 0.05$ | $5.03 \pm 0.31$ | $16.48 \pm 0.08$ | $\mathbf{259.50 \pm 0.32}$ |
| $\alpha_*$-VAE | $\mathbf{8.52 \pm 0.07}$ | $\mathbf{9.59 \pm 0.03}$ | $\mathbf{3.88 \pm 0.27}$ | $\mathbf{15.98 \pm 0.17}$ | $259.52 \pm 0.36$ |

*Table 1.* Final model reconstruction error across regularisation divergences and datasets. For trainable skew $\alpha$-VAEs (bottom two rows), final $\alpha$ values are given in Table 3. $\alpha_*$ indicates dual $\alpha$-VAEs.

plots delineate data and encoder dependency, with consistency across seeds which initialise model parameters and training procedures (e.g. dropout and batching), suggesting learnt skew predominantly derives from, and varies between, datasets (see Table 3).

**Robustness to noise.** As learning skew allows for problem-dependent measurement of the distance to the prior, which better accommodates more dispersed encoded distributions, we tested how learning skew regularises VAEs in various noise settings. In Figure 5, we present denoising experiments where we add Gaussian noise, $\mathcal{N}(0, \sigma^2)$, to the normalised input images and clip to $[0, 1]$. Despite degraded intuition surrounding where the skew balance should lie in the noisy setting, Figure 5 clearly demonstrates that our trainable skew in $\mathrm{JS}^{\mathrm{G}_\alpha}$, and its dual form, consistently provides lower test set reconstruction loss across noise levels. As a sanity check, we can also verify expected trends in the other divergences, with the less dispersion-friendly reverse KL performing poorly at higher noise levels as forward KL becomes more appropriate, as well as the expected robust behaviour of MMD at high noise levels (Zhao et al., 2019).

**Improved reconstruction as a rate-distortion trade-off.** We test our model's reconstruction-loss performance in the standard clean-image setting. Table 1 demonstrates superior reconstruction loss across multiple datasets, outperforming both the naive choice of fixed $\alpha = 0.5$ and KL divergence—the latter by a substantial margin. We further detail the relationship with fixed $\alpha$ in Figure 1a, Figure 6, the supplementary figures in Appendix D, observing low reconstruction loss from an arbitrary starting skew. To explain this performance gain, we plot distortion (MSE) against rate (Figure 7 in Appendix D), measured using the reverse KL divergence for all divergences. The shift down and to the right for $\alpha$-VAEs mean that they trade off rate for distortion, even more so when skew is learnt, giving higher quality image reconstruction when starting with unknown divergence scaling $\lambda$.

## 5. Conclusion

We recast VAE optimisation as a multi-objective task with competing reconstruction and divergence regularisation terms. This allowed us to use gradient descent to directly learn the divergence skew used to control the variational distribution $q_\phi(z|x)$ relative to the prior distribution $p(z)$. The resulting method, $\alpha$-VAE, was shown to be well-posed as a 1D optimisation with a dependency on both the encoder and the data, has improved reconstruction and denoising properties over standard regularisation techniques, and avoids an expensive grid search over skew values. Moreover, as $\alpha$-VAEs also generalise over the $\beta$-VAE family, they are a practical, efficient, and unbiased choice of VAE across a range of tasks and domains.

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

## A. Related work

Since its introduction (Nielsen, 2021), $\text{JS}^{G_\alpha}$ (with $\alpha = 0.5$) has been used to decompose and estimate multi-modal ELBO loss as regularisation in multi-modal VAEs. (Deasy et al., 2020) further investigated the potential of $\text{JS}^{G_\alpha}$ at different skew values, in turn, demonstrating improved reconstruction performance of VAEs. In contrast, in this paper we propose data-dependent learnable skew, which retains the benefits of using an optimal skew but avoids an expensive grid-search. This provides both efficient and practical way of training VAEs, leading to improved denoising and reconstruction performance as well as better lossy-compression rates focussed on in (Huang et al., 2020). In a similar context, our work is related to (Brekelmans et al., 2020), that introduces an optimal integration schedule via dynamic parameter selection when approximating the Thermodynamic Variational Objective (Masrani et al., 2019).

$\alpha$-VAEs extend the standard VAEs (Kingma & Welling, 2014; Rezende et al., 2014) paradigm with regularisation constraint inspired by recent work on closed-form expressions for statistical divergences (Nielsen, 2019; Nishiyama, 2018). In particular, $\alpha$-VAEs offer a stable and intuitive regularisation mechanism. This allows optimal interpolation between forward and reverse KL divergence, therefore combating the issue of posterior collapse (Lucas et al., 2019). In this regard, our work is related to approaches that address this issue through KL annealing during training (Bozkurt et al., 2021; Huang et al., 2018; Burgess et al., 2018). In a more general sense, this work is also related to other approaches that utilise various statistical divergences and distances for latent space regularisation as an alternative to the conventional KL divergence (Hensman et al., 2014; Tolstikhin et al., 2018; Dieng et al., 2017; Zhang et al., 2019; Zhao et al., 2019; Li & Turner, 2016).

## B. Proofs

### B.1. Proposition 1.

*Proof.* First, simplifying the divergence

$$
\begin{aligned}
\text{JS}^{G_\alpha} &= (1-\alpha)\text{KL}\left(p \parallel p^\alpha q^{1-\alpha}\right) + \alpha\text{KL}\left(q \parallel p^\alpha q^{1-\alpha}\right) \\
&= (1-\alpha)\int_x p\log\left[\frac{p}{p^\alpha q^{1-\alpha}}\right]dx \\
&\quad + \alpha\int_x q\log\left[\frac{q}{p^\alpha q^{1-\alpha}}\right]dx \\
&= (1-\alpha)^2\int_x p\log\left[\frac{p}{q}\right]dx + \alpha^2\int_x q\log\left[\frac{q}{p}\right]dx \\
&= (1-\alpha)^2\text{KL}(p \parallel q) + \alpha^2\text{KL}(q \parallel p).
\end{aligned} \tag{18}
$$

Then, differentiating (18) with respect to $\alpha$

$$
\frac{d\text{JS}^{G_\alpha}}{d\alpha} = 2((\alpha-1)\text{KL}(p \parallel q) + \alpha\text{KL}(q \parallel p)) = 0, \tag{19}
$$

and rearranging gives Equation (16), before differentiating again

$$
\frac{d^2\text{JS}^{G_\alpha}}{d\alpha^2} = 2(\text{KL}(p \parallel q) + \text{KL}(q \parallel p)) \geq 0, \tag{20}
$$

demonstrates the global minimum property as KL divergence is always positive. $\square$

### B.2. Proposition 2.

*Proof.* We first define the convexity strength and smoothness of $f(\alpha) = \text{JS}^{G_\alpha}$, using its simplified form in Equation (18).

A function $f(\alpha) : \mathbb{R}^n \mapsto \mathbb{R}^n$ is $\lambda$-strongly convex if

$$
\nabla^2 f(\alpha) \succeq \lambda I, \tag{21}
$$

where $\succeq$ is a generalised inequality and, in our 1D case, this expression reduces to the second derivative with respect to $\alpha$, giving

$$
\lambda = 2(\text{KL}(p \parallel q) + \text{KL}(q \parallel p)). \tag{22}
$$

$f(\alpha)$ is also $\beta$-smooth if $\frac{df}{d\alpha}$ is $\beta$-Lipschitz

$$
||\nabla f(x) - \nabla f(y)|| \leq \beta||x-y||, \tag{23}
$$

and, as our $f$ is doubly differentiable, the Mean Value Theorem (Rudin et al., 1976) with $g = \frac{df}{d\alpha}$

$$
\frac{g(x)-g(y)}{x-y} \leq \frac{dg(z)}{d\alpha} = \frac{d^2 f(z)}{d\alpha^2} \quad \forall x < z < y, \tag{24}
$$

gives the necessary bound and the same constant

$$
\beta = 2(\text{KL}(p \parallel q) + \text{KL}(q \parallel p)). \tag{25}
$$

We can now substitute into the result from convex optimisation with gradient descent (Bubeck, 2014) that, for a $\lambda$-strongly convex and $\beta$-smooth function, the optimal step size $\gamma$ is

$$
\gamma = \frac{2}{\lambda + \beta} \tag{26}
$$

$$
= \frac{1}{2(\text{KL}(p \parallel q) + \text{KL}(q \parallel p))}, \tag{27}
$$

and the upper bound on the convergence rate is

$$
\beta||\alpha_1 - \alpha^*||_2^2 e^{-\frac{4T}{\kappa}}, \tag{28}
$$

where $\kappa = \frac{\beta}{\lambda} = 1$ is the condition number, completing the proof. $\square$

| Dataset | Stage | Architecture |
|---|---|---|
| MNIST | Input | 28x28x1 zero padded to 32x32x1. |
| | Encoder | Repeat Conv 32x4x4 for 3 layers (stride 2, padding 1). FC 256, FC 256. ReLU activation. |
| | Latents | 10. |
| | Decoder | FC 256, FC 256, Repeat Deconv 32x4x4 for 3 layers (stride 2, padding 1). ReLU activation, Sigmoid. MSE. |
| Fashion-MNIST | Input | 28x28x1 zero padded to 32x32x1. |
| | Encoder | Repeat Conv 32x4x4 for 3 layers (stride 2, padding 1). FC 256, FC 256. ReLU activation. |
| | Latents | 10. |
| | Decoder | FC 256, FC 256, Repeat Deconv 32x4x4 for 3 layers (stride 2, padding 1). ReLU activation, Sigmoid. Bernoulli. |
| dSprites | Input | 64x64x1. |
| | Encoder | Repeat Conv 32x4x4 for 4 layers (stride 2, padding 1). FC 256, FC 256. ReLU activation. |
| | Latents | 10. |
| | Decoder | FC 256, FC 256, Repeat Deconv 32x4x4 for 4 layers (stride 2, padding 1). ReLU activation, Sigmoid. Bernoulli. |
| Chairs | Input | 64x64x1. |
| | Encoder | Repeat Conv 32x4x4 for 4 layers (stride 2, padding 1). FC 256, FC 256. ReLU activation. |
| | Latents | 32. |
| | Decoder | FC 256, FC 256, Repeat Deconv 32x4x4 for 4 layers (stride 2, padding 1). ReLU activation, Sigmoid. Bernoulli. |

*Table 2.* Detail of model architectures.

## C. Benchmark datasets for skew exploration

Throughout our experiments we evaluate the reconstruction loss (mean squared error) on four standard benchmark image datasets : MNIST, $28 \times 28$ black and white images of handwritten digits (LeCun et al., 2010); Fashion-MNIST, $28 \times 28$ black and white images of clothing (Xiao et al., 2017); Chairs, $64 \times 64$ black and white images of 3D chairs (Aubry et al., 2014); dSprites $64 \times 64$ black and white images of 2D shapes procedurally generated from 6 ground truth independent latent factors (Matthey et al., 2017); CelebA resampled to $64 \times 64 \times 3$ colour images of celebrity faces (Liu et al., 2015). For fair comparison, we follow Higgins et al. (2017) by selecting a common neural architecture across experiments. For consistent analysis, rather than searching within architecture and hyperparameter spaces for the best performing model by some metric, we standardise comparison and characterise the benefit of learning divergence skew.

In terms of model details, we use the architectures specified in Table 2 throughout experiments. We pad 28x28x1 images to 32x32x1 with zeros as we found resizing images negatively affected performance, and we use nearest neighbour interpolation to downsample CelebA to be 64x64x3. We use a learning rate of 1e-4 throughout and use batch size 64 and 256 for the two MNIST variants and the other datasets respectively. Where not specified (e.g. momentum coefficients in Adam (Kingma & Ba, 2014)), we use the default values from PyTorch (Paszke et al., 2019). The only architectural change we make between datasets is an additional convolutional (and transpose convolutional) layer for encoding (and decoding) when inputs are 64x64x$N$ instead of 32x32x1. We train dSprites and CelebA for 50 epochs, and all other datasets for 100 epochs. All models were trained on one GPU, with type varying between: NVIDIA Titan X, NVIDIA 2080 Ti, or NVIDIA A100.

# D. Further results

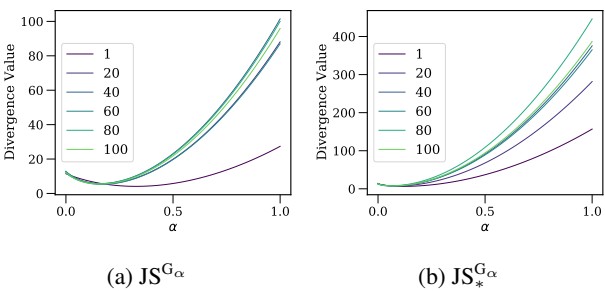

(a) $JS^{G_\alpha}$.  (b) $JS_*^{G_\alpha}$.

*Figure 3.* $\alpha$ landscape at different training epochs in training for FashionMNIST.

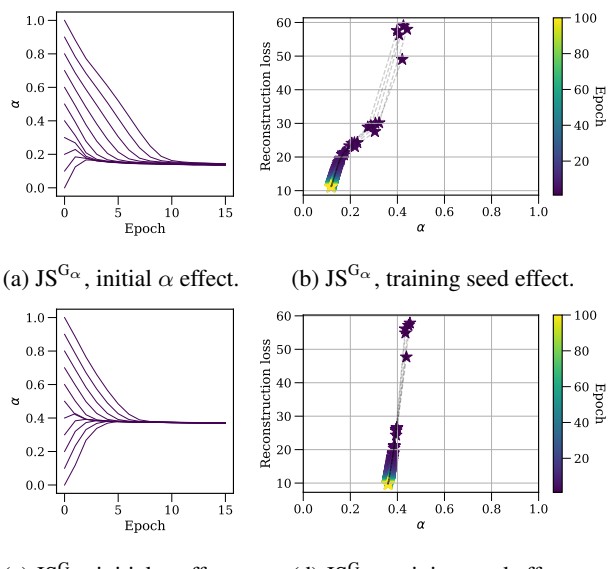

(a) $JS^{G_\alpha}$, initial $\alpha$ effect.  (b) $JS^{G_\alpha}$, training seed effect.

(c) $JS_*^{G_\alpha}$, initial $\alpha$ effect.  (d) $JS_*^{G_\alpha}$, training seed effect.

*Figure 4.* The robust nature of our method's convergence. $\alpha$ convergence across a range of starting values and seeds using the MNIST dataset.

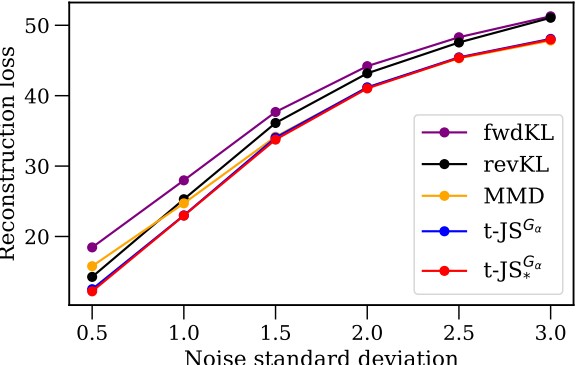

*Figure 5.* Reconstruction loss for noisy input images across different noise levels and regularisation divergences on MNIST.

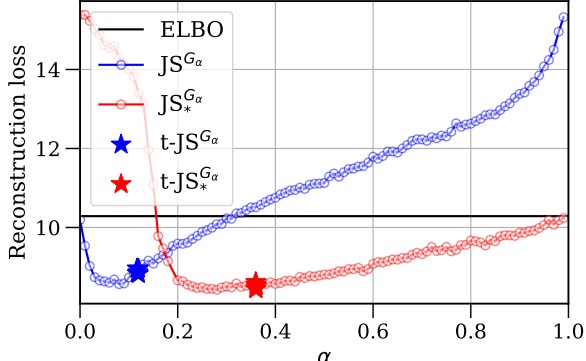

*Figure 6.* $\alpha$-VAEs on the Fashion-MNIST dataset.

| Divergence | MNIST | FashionMNIST | dSprites | Chairs | CelebA |
|---|---|---|---|---|---|
| t-JS$^{G_\alpha}$ | $0.118 \pm 0.001$ | $0.165 \pm 0.002$ | $0.06 \pm 0.007$ | $0.121 \pm 0.001$ | $0.0377 \pm 0.000$ |
| t-JS$_*^{G_\alpha}$ | $0.360 \pm 0.000$ | $0.377 \pm 0.002$ | $0.39 \pm 0.001$ | $0.365 \pm 0.001$ | $0.310 \pm 0.000$ |

*Table 3.* Mean learnt $\alpha$ for JS$^{G_\alpha}$ and JS$_*^{G_\alpha}$ with standard deviation across 5 different training seeds.

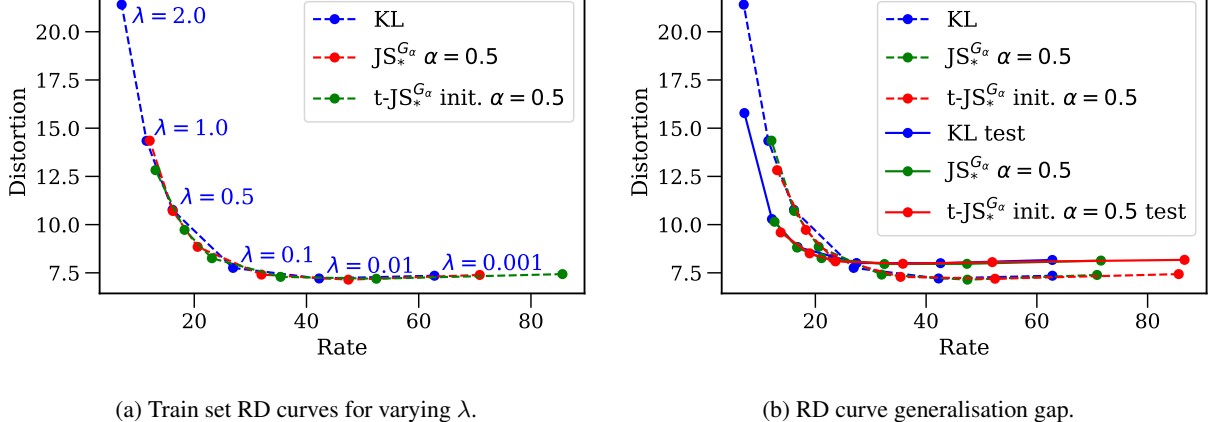

(a) Train set RD curves for varying $\lambda$.

(b) RD curve generalisation gap.

*Figure 7.* Rate distortion curves for KL, JS$_*^{G_\alpha}$, and JS$_*^{G_\alpha}$ with learnt $\alpha$ on MNIST. Dashed or full lines connect values from the training or test set respectively. $\lambda$ values vary consistently for each divergence from left to right but are only annotated for KL.