# OpenReview forum: "$\alpha$-VAEs : Optimising variational inference by learning data-dependent divergence skew"
_ICML.cc/2021/Workshop/INNF — INNF+ 2021 poster_

### Official Review · Reviewer_K7LH · 2021-06-07

**Rating:** Borderline Accept
**Confidence:** 4

**Summary:**

The paper proposes to train VAEs using a skew-geometric Jensen-Shannon divergence. Different from previous work is that the skew parameter $\alpha$ in the divergence is learned from data. It is shown that the optimal $\alpha$ depends on the data and the encoder and that it does not tend to the extreme values $\alpha=0$ or $\alpha=1$. It is shown that $\alpha$-VAE yield lower reconstruction errors compared to vanilla VAEs, beta-VAEs, InfoVAEs and $\alpha$-VAE with $\alpha=0.5$.

**Justification For Rating:**

The idea is clearly explained and evaluated.
I have some concerns about the approach of training parameters on the objective function itself on data.
However, since this is a workshop I'm leaning towards acceptance such that authors can discuss and get feedback on their work.

---

### Official Review · Reviewer_wqNr · 2021-06-13

**Rating:** Borderline Accept
**Confidence:** 4

**Summary:**

This paper proposes to use the skew-geometric Jensen-Shannon Divergence (JSD) -- with learned divergence parameter $\alpha$ -- to optimize the divergence-based regularization term in variational auto-encoders (VAEs). Previous work has attempted to find $\alpha$ via expensive grid search, but this automatic approach shows promise in a variety of practical domains.

**Justification For Rating:**

I think this paper is very polished for a workshop submission, both in terms of writing and experimental results. However, my main criticism is that this paper does not actually include anything on normalizing flows, which makes me somewhat concerned about its fit in the workshop. Nevertheless, I've voted to include it because of the focus on likelihood models, despite it being an approximate likelihood model.

I had some additional comments/questions about the paper:
- How exactly is the optimization for $\alpha$ and the other parameters alternated? Since the optimization is marginally convex in $\alpha$, do we find an optimal value at each epoch?
- The results seem focused on reconstruction error. Is it possibly we are overfitting to this metric? How about log-likelihood?
- Dual $\alpha$-VAEs are mentioned a fair bit in the experiments section, but not defined in the manuscript.
- Should be "maximise" the ELBO L44, left
- If you are using natbib for citations, you may want to use \citet at times for papers referred to as a noun in a sentence, e.g. L74 left column

---

### Official Review · Reviewer_uLtY · 2021-06-14

**Rating:** Reject
**Confidence:** 4

**Summary:**

The vanilla VAE uses KL divergence to regularizer the posterior to the prior. Recently, the skew-geometric JS diverence ($\text{JS}^{G_\alpha}$) with a skew parameter $\alpha$ was introduced as an improvement over the KL divergence, but it required searching over the values of $\alpha$. This paper proposes to treat $\alpha$ as a model parameter and optimize it together with all other parameters. The authors also show that $\text{JS}^{G_\alpha}$ is convex is $\alpha$ and admits a closed-form solution when the functional is evaluated on Gaussian distributions. A toy domain evaluation shows clear trends of optimal $\alpha$ with respect to problem multi-modality and dimensionality. Additionally, optimizing $\alpha$ leads to lower reconstruction errors in image domains.

**Justification For Rating:**

The paper is fairly well written (comments below), and the presented analysis makes sense. However, this work is very incremental (simply performing gradient-descent on a parameter of another algorithm, with respect to which that algorithm is trivially differentiable). Additionally, I think the presented evaluation is incorrect and can be very misleading.
- Using a different divergence than KL in the optimization objective means that the objective is not a valid lower bound anymore. This leads to the loss of interpretability of what it is that we are optimizing. That would be fine if the optimized objective still lead to a higher value of marginal data log-likelihood. However, the paper does not evaluate the marginal, nor does it even mention that the optimized objective is not a valid lower-bound.
- The only argument in favour of the presented approach is that of lower reconstruction loss when using optimized $\alpha$. This makes sense, since optimal $\alpha$ leads to a lower value of the divergence and probably a weaker regularizer. This gives more freedom to the posterior distribution and leads to lower MSE. While this is fine, we generally do not need optimization to weaken the regularizer: $\beta$-VAE with $\beta < 1$ is enough, or just dropping the KL altogether. This paper, however, tries to make an argument that optimizing $\alpha$ leads to "better" results without specifying what "better" means. I would buy that argument if the authors showed improved marginal likelihoods as a result of optimizing $\alpha$, but looking at the rate-distortion tradeoff in Fig. 7 this is probably not the case.

Comments on writing/clarity:
- end of paragraph 1 in sec 1: prior distribution in VAE need not be fixed.
- the first two sentences of par 2 in sec 1 make no sense.
- citing a whole book (in par 3, sec 1) is not very helpful; please cite specific section/page.
- In sec 2.1 you cite $\beta$-VAE of Higgins et. al. wrt constrained optimization. I think you meant to cite GECO of Rezende et. al. instead?

---

### Decision · Program_Chairs · 2021-06-15

**Decision:**

Accept (poster)

**Comment:**

The topic of this paper is related to the theme of the workshop and the paper has received mixed reviews. The program chairs discussed whether or not evaluating only with reconstruction error is sufficient. We came to the conclusion that such discussions are perhaps worth having openly at the workshop. We have therefore decided to accept this paper. Please take into account the suggestions and comments by the reviewers when submitting the camera ready version.